# Ultrasound pupillometry for the detection of a relative afferent pupillary defect (RAPD): Systematic evaluation in patients with optic neuritis and comparison with infrared video pupillometry

**Franziska Siebald[1], Ulrike Grittner[2], Carolin Otto[1], Charlotte Bereuter[3,4], Hanna G. Zimmermann[3,4], Lutz Harms[1], Jan Klonner[5], Stephan J. Schreiber[6], Friedemann Paul[1,3,4], Klemens Ruprecht[1☯], Felix A. Schmidt[1,7,8,9☯]***

1 Department of Neurology, Charité – Universitätsmedizin Berlin, Corporate Member of Freie Universität Berlin and Humboldt-Universität zu Berlin, Berlin, Germany, 2 Institute of Biometry and Clinical Epidemiology, Charité – Universitätsmedizin Berlin, Corporate Member of Freie Universität Berlin and Humboldt-Universität zu Berlin, Berlin, Germany, 3 NeuroScience Clinical Research Center (NCRC), Charité – Universitätsmedizin Berlin, Corporate Member of Freie Universität Berlin and Humboldt-Universität zu Berlin, Berlin, Germany, 4 Experimental and Clinical Research Center (ECRC), Charité – Universitätsmedizin Berlin, Corporate Member of Freie Universität Berlin and Humboldt-Universität zu Berlin, Berlin, Germany, 5 Department of Ophthalmology, Charité – Universitätsmedizin Berlin, Corporate Member of Freie Universität Berlin and Humboldt-Universität zu Berlin, Berlin, Germany, 6 Department of Neurology, Klinik Hennigsdorf, Oberhavel Kliniken GmbH, Hennigsdorf, Germany, 7 Berlin Institute of Health (BIH), Berlin, Germany, 8 Medical School Berlin (MSB) Hochschule für Gesundheit und Medizin, Berlin, Germany, 9 Department of Neurology, Michels Kliniken - Brandenburgklinik Berlin-Brandenburg, Bernau bei Berlin, Germany

☯ These authors contributed equally to this work.
* felix.schmidt@brandenburgklinik.de

## Abstract

### Purpose

A relative afferent pupillary defect (RAPD) is a characteristic clinical sign of optic neuritis (ON). Here, we systematically evaluated ultrasound pupillometry (UP) for the detection of an RAPD in patients with ON, including a comparison with infrared video pupillometry (IVP), the gold standard for objective pupillometry.

### Materials and methods

We enrolled 40 patients with acute (n = 9) or past (n = 31) ON (ON+), 31 patients with multiple sclerosis (MS) without prior ON, and 50 healthy controls (HC) in a cross-sectional observational study. Examinations comprised the swinging flashlight test, B-mode UP, IVP, autorefraction to assess the best-corrected visual acuity, optical coherence tomography to determine peripapillary retinal nerve fiber layer thickness, and the 51-item National Eye Institute-Visual Function Questionnaire to determine the vision-related quality of life.

**Data Availability Statement:** All relevant data are within the manuscript and its Supporting information files, see document 'Data Table' uploaded as Supporting information item.

**Funding:** Felix A. Schmidt received funding from the Berlin Institute of Health (https://www.bihealth.org/) and the Stiftung Charité, Berlin, Germany (https://www.stiftung-charite.de/) in the Clinician Scientist programm. Klemens Ruprecht was a participant in the BIH Clinical Fellow Program funded by Stiftung Charité, Berlin, Germany. None of the funding organizations had a role in the design or conduct of this study".

**Competing interests:** I have read the journal's policy and the authors of this manuscript have the following competing interests: Felix A. Schmidt received support from the Berlin Institute of Health (BIH) and the Stiftung Charité, Berlin, Germany (Clinician Scientist Grant). Klemens Ruprecht received research support from Novartis, Basel, Switzerland, Merck Serono, German Ministry of Education and Research, Stiftung Charité (BIH Clinical Fellow Program), Germany, European Union (821283-2), Guthy Jackson Charitable Foundation, Beverly Hills, CA, USA, and Arthur Arnstein Foundation, Berlin, Germany; received speaker honoraria from Novartis and Virion Serion, Würzburg, Germany, and travel grants from Guthy Jackson Charitable Foundation. All other authors have declared that no competing interests exist. This does not alter our adherence to PLOS ONE policies on sharing data and materials.

## Results

While UP and IVP measurements of pupil diameter (PD) at rest correlated in ON+ eyes (n = 52, $r$ = 0.56, 95% CI: 0.35; 0.72) and in HC eyes (n = 100, $r$ = 0.60, 95% CI: 0.47; 0.72), PD at rest was smaller in UP than in IVP measurements (difference, mean (SD) ON+ eyes: 0.44 (0.87) mm, HC eyes: 0.69 (0.80) mm). RAPD assessment by UP sharply discriminated acute ON eyes (n = 9) and HC eyes (n = 100, AUC = 1, 95%CI: 1; 1). UP detected an RAPD in 5/31 (16%) patients with MS without prior ON who had not exhibited an RAPD during the swinging flashlight test. In ON+ eyes (n = 52), UP showed stronger correlations with visual acuity ($r$ = 0.66, 95% CI: 0.50; 0.78) and vision-related quality of life ($r$ = 0.47, 95% CI: 0.24; 0.66) than IVP ($r$ = 0.52, 95% CI: 0.36; 0.67 and $r$ = 0.27, 95% CI: 0.03; 0.51).

## Conclusions

B-mode UP allows for objective detection and quantification of an RAPD with performance characteristics similar to IVP. RAPD assessment by UP may detect subclinical optic nerve damage in patients with MS. We propose a standardized protocol for RAPD detection by UP that can be used in routine clinical evaluation of patients with ON or other optic neuropathies.

## Introduction

Optic neuritis (ON) is a common manifestation of inflammatory diseases of the central nervous system, such as multiple sclerosis (MS), neuromyelitis optica spectrum disorder (NMOSD), or myelin oligodendrocyte glycoprotein antibody associated disease and can result in substantial visual impairment and diminished quality of life [1–3]. Clinical symptoms of ON include subacute unilateral vision loss, blurred vision, pain during eye movement, and loss of color vision [4]. A characteristic clinical sign of unilateral ON is a relative afferent pupillary defect (RAPD), defined as an asymmetric pupillary response to identical light stimuli [4, 5]. An RAPD can be evaluated at the bedside by the swinging flashlight test (SFLT), which involves observing the pupillary light reflex (PLR) while swinging a penlight from the healthy to the affected eye and vice versa [6]. In case of a unilateral ON, a direct light stimulus ($L_{stim}$) of the affected eye results in a reduced and slowed constriction or even paradoxical dilatation of both pupils, referred to as RAPD. In contrast, after a consensual $L_{stim}$ of the contralateral unaffected eye, both pupils constrict simultaneously and equally [7]. However, accurate and reproducible assessment of an RAPD by the SFLT depends on the examiner's expertise and can be impeded by lighting conditions, dark eye color, or limited eyelid retraction [8].

We previously developed a strategy to objectively assess the PLR using B-mode ultrasound pupillometry (UP) and showed that UP can be used to detect an RAPD [9, 10]. In UP, patients are examined with their eyes closed to prevent contact of the ultrasound gel with the eye, while the examiner uses an external manual light source (penlight).

Infrared video pupillometry (IVP), the current gold standard for objective assessment of the PLR, provides precise quantitative measurements and allows hereby for the analysis of the PLR through high-resolution video imaging [11]. For IVP, the infrared video camera records the PLR of the ipsilateral open eye after an $L_{stim}$ of the integrated light diode with a fixed distance between the eye and the light source. IVP has previously been applied for the detection of an RAPD in patients with ON where it was shown to detect an RAPD with high sensitivity

and specificity [12, 13]. However, unlike ultrasound, IVP devices are rarely available in non-specialized clinical settings. Furthermore, IVP assessment requires the ability to fixate on a target and to keep the eyes open throughout the examination, whereas UP does not require active patient cooperation, provided that the patient is able to remain seated or lie quietly during the procedure. A recent study found a strong correlation (Pearson's correlation coefficient, $r = 0.831$, $p < 0.01$) between UP- and IVP-based PLR assessments in healthy subjects [14]. However, comparative studies of UP and IVP for the assessment of RAPD in patients with ON have hitherto not been performed.

In this exploratory study, we systematically evaluated UP for the detection of an RAPD in patients with ON, including a comparison with IVP. We also investigated whether UP might be more sensitive than the SFLT for the detection of subclinical optic nerve damage in patients with MS. Finally, we put results of UP and IVP into perspective with other visual endpoints of ON, namely visual acuity (VA), absolute peripapillary retinal nerve fiber layer thickness (pRNFLT), and vision-related quality of life (vQoL).

## Materials and methods

### Study design

This cross-sectional observational study was approved by the Institutional Review Board of Charité –Universitätsmedizin Berlin (EA1/190/15) and was conducted in accordance with the current applicable ethical guidelines of the Declaration of Helsinki and German law. Written informed consent was obtained from all participants prior to inclusion into the study and patient confidentiality was ensured throughout the study.

### Study participants

Patients with acute or past ON were recruited from the Departments of Neurology and Ophthalmology at Charité –Universitätsmedizin Berlin, Berlin, Germany, between September 24th, 2019 and December 9th, 2020. We defined acute ON as symptom onset <30 days and past ON as symptom onset ≥30 days before the examination day. Patients with more than one episode of ON were also included. We also recruited patients with a diagnosis of MS according to the McDonald 2017 criteria [15] who had no clinical history of ON (MS ON-). Healthy controls (HC), with sex and age distribution comparable to the patient groups, were recruited from hospital staff or medical students. Participants had to be ≥18 and ≤70 years of age.

Exclusion criteria were ophthalmologic diseases other than ON, such as amblyopia, significant anisocoria, efferent defects, use of medication known to affect pupillary function, recent ocular surgery, or optical laser treatment <6 months before the examination date, and an Expanded Disability Status Scale (EDSS) score ≥6 in patients with MS or NMOSD [16].

All participants underwent a standardized examination procedure in the order described below. All tests were performed by the same examiner (FrS).

### Swinging flashlight test

The examination room was darkened during SFLT, UP, and IVP testing. Adequate adaptation time was allowed before and between the different tests. SFLT was assessed with a standard penlight (70,000 lux).

### Ultrasound pupillometry

B-mode UP was performed with an Esaote Mylab 25 system (Esaote, Genova, Italy) with a 10 MHz linear probe, as previously described in detail [9, 10]. Participants were examined in the

recumbent position with the eyes closed and the probe initially placed on the left lower eyelid, focused on the pupil. We performed SFLT with the same penlight as for bedside SFLT, with an approximative distance of 2.5 cm between the eye and the penlight, while recording a 5-second video of the left PLR. After 2 minutes of recovery, the right eye was examined in the same manner. Pupil diameter (PD) at rest and PD after direct and consensual $L_{stim}$ were assessed with the built-in measuring device. To facilitate the introduction of UP into routine clinical practice, we developed a short clinical examination protocol, which we added as a separate section after the discussion.

### Infrared video pupillometry

We used an NPi-200 pupillometer (NeurOptics, Irvine, CA, USA). After an $L_{stim}$ of 800 milliseconds (ms) with a light diode (1,100 lux), an integrated infrared camera recorded the ipsilateral PLR with a fixed distance of 2.5 cm between the eye and the light source. We evaluated PD at rest and PD after direct $L_{stim}$ for both eyes.

### Visual acuity

Best-corrected monocular VA was assessed with an AR-1s autorefractor (Oculus/Nidek, Wetzlar, Germany). The optotype charts with letters and numbers allowed for VA assessment from 0.1 to 1.25 dec in logarithmic steps. We considered VA = 0 if the participant could not perceive 0.1 dec.

### Peripapillary retinal nerve fiber layer thickness

We evaluated the absolute pRNFLT by retinal optical coherence tomography (OCT) with a CIRRUS HD-OCT (Carl Zeiss Meditec AG, Jena, Germany) using Optic Disc Cube 200x200 scans. We calculated the intereye absolute difference (IEAD) in µm and the intereye percentage difference (IEPD) in percent of both eyes of an individual. We considered IEAD≥5 µm and IEPD≥5% as indicative of a history of unilateral ON in patients with MS or NMOSD with onset of the last ON episode ≥3 months before the examination day [17, 18].

### Vision-related quality of life

We assessed vQoL using the German adaptation [19] of the National Eye Institute-Visual Function Questionnaire (NEI-VFQ) [20]. The results of 51 questions with 12 subscales were averaged to obtain a final score of vQoL ranging from 0 to 100 [21]. The NEI-VFQ is a reliable and valid tool to assess vQoL in a variety of chronic eye conditions. Its strong psychometric properties make it suitable for group-level comparisons in clinical research, offering precise and consistent evaluation of how visual impairment affects the patients' daily life [22]. To provide an understanding of the items included in the NEI-VFQ, we here present a sample question and the Likert scale of the possible responses. Question: How much difficulty do you have reading street signs or the names of stores? Answers: No difficulty at all, a little difficulty, moderate difficulty, extreme difficulty, stopped doing this because of your eyesight, stopped doing this for other reasons or not interested in doing this [20].

### Analysis of pupillometry data

The constriction amplitude was defined as the difference between PD at rest and PD after $L_{stim}$ (direct or consensual). The constriction ratio (CR) is the quotient of the consensual and direct constriction amplitudes and is a quantitative measure of an RAPD [10]. We previously

proposed CR = 1.3 as a threshold to differentiate between eyes with (CR>1.3) and without (CR≤1.3) RAPD [10].

The constriction range in percent describes the degree of pupil constriction after $L_{stim}$, calculated as follows:

$$\text{Constriction range} = \left( 1 - \frac{\text{PD after Lstim}}{\text{PD at rest}} \right) \times 100$$

Pupil constriction time (PCT) is the time in ms between an $L_{stim}$ and maximum miosis and was manually evaluated using AVSVideoConverter12.1 freeware (Online Media Technologies Ltd., London, UK) on recorded video sequences of UP [9].

## Statistics

To compare UP and IVP, we calculated differences, means, and standard deviations (SD). We evaluated the correlation ($r$) between the two methods by using the square root of the marginal $r^2$ values based on simple linear mixed models that account for repeated measures within individuals (right and left eye) [23, 24]. Similarly, we calculated the correlation of PLR assessments with VA, pRNFLT, and vQoL. We visualized individual differences in PD in UP and IVP using a Bland-Altman plot to assess possible systematic deviations. We determined the area under the curve (AUC) by receiver operating characteristic (ROC) curves to assess the diagnostic accuracy of the CR method in UP to detect an RAPD by using binary logistic regression models with random intercepts to account for repeated measures within individuals. To analyze mean differences between study groups and HC in PCT, VA, and pRNFLT, we used linear mixed models with random intercepts for participants to account for repeated measures within individuals. These models were additionally adjusted for age. For internal reliability analysis of the vQoL, we calculated Cronbach's alpha to assess the internal reliability of the NEI-FVQ. We used one-factor analysis of variance (ANOVA) followed by Dunnett's test to compare IEAD, IEPD, and vQoL of the study groups with HC. Except for ANOVA, results were not adjusted for multiple testing. A two-sided significance level of α = 0.05 was considered. *P*-values (*p*) should be interpreted with caution in this exploratory study. Interpretation of results was based on effect estimates, 95% confidence intervals (95% CI) and effect size measures. We performed statistical analyses and graphs with SPSS 27 (IBM SPSS Statistics, Armonk, NY, USA) and R version 4.3.0 (R Foundation for Statistical Computing, Vienna, Austria) [25].

## Results

All data used in the study are fully documented and available in S1 Table in the Supporting information.

## Participants

In this study, we included 9 patients with acute ON, 31 patients with past ON (collectively referred to as ON+ patients), 31 MS ON- patients, and 50 HC.

All 9 patients with acute ON had unilateral ON (acute ON eyes, n = 9). In 4/9 (44%) of these patients, this was the first episode of ON, and 5/9 (56%) had experienced previous episodes of ON (3 patients with one and 2 with three previous episodes of ON in the currently affected eye, respectively). In addition, 2/9 (22%) patients had experienced previous ON episodes in the contralateral (currently unaffected) eye.

In patients with past ON (n = 31), the mean (SD) time since last ON was 81.7 (86.7) months with a minimum of 3 months. Of these, 5/31 (16%) patients had bilateral ON with an equal

**Table 1. Demographic and clinical characteristics of the study participants.**

| Study group | | acute ON (n = 9) | past ON (n = 31) | MS ON- (n = 31) | HC (n = 50) |
|---|---|---|---|---|---|
| Sex | male/female, n (% female) | 4/5 (56) | 7/24 (77) | 13/18 (58) | 20/30 (60) |
| Age | years, mean (SD) | 34 (11) | 42 (11) | 39 (12) | 44 (14) |
| Ethnicity | Asian, n (%) | 1 (11) | 2 (7) | 1 (3) | 2 (4) |
| | Black, n (%) | 0 (0) | 0 (0) | 1 (3) | 1 (2) |
| | White, n (%) | 8 (89) | 29 (93) | 29 (94) | 47 (94) |
| Diagnosis | CIS, n (%) | 1 (11) | 2 (7) | 0 (0) | n.a. |
| | NMOSD, n (%) | 4 (44) | 1 (3) | 0 (0) | n.a. |
| | PPMS, n (%) | 0 (0) | 0 (0) | 3 (10) | n.a. |
| | RRMS, n (%) | 4 (44) | 27 (87) | 28 (90) | n.a. |
| | SPMS, n (%) | 0 (0) | 1 (3) | 0 (0) | n.a. |
| EDSS | median (Q1-Q3) | 3.0 (1.8–4.0) | 2.0 (1.5–3.0) | 1.5 (1.0–2.3) | n.a. |
| Time since last ON | months, mean (SD) | 0.4 (0.3) | 81.7 (86.7) | n.a. | n.a. |

CIS = clinically isolated syndrome, EDSS = Expanded Disability Status Scale, HC = healthy controls, ON = optic neuritis, MS ON- = patients with multiple sclerosis without a history of ON, n.a. = not applicable, NMOSD = neuromyelitis optica spectrum disorder, PPMS = primary progressive multiple sclerosis, Q1/3 = first/third quartile, RRMS = relapsing remitting multiple sclerosis, SD = standard deviation, SPMS = secondary progressive multiple sclerosis.

number of ON episodes in both eyes. Specifically, 1/5 had two bilateral simultaneous ON episodes and 4/5 had bilateral sequential ON episodes, with one ON episode per eye. In addition, 5/31 (16%) had past bilateral ON but with an unequal number of ON episodes in both eyes, and 21/31 (68%) had past unilateral ON, resulting in a total number of n = 43 eyes with past ON. Altogether, there were n = 52 eyes with acute or past ON (ON+ eyes) in 40 individuals.

Unaffected eyes of patients with acute (n = 7) and past ON (n = 21), all eyes of MS ON-patients (n = 62) and HC (n = 100) were defined as ON- eyes (total n = 190).

The demographic and clinical characteristics of the study participants are shown in Table 1.

## Prevalence of an RAPD as assessed by the swinging flashlight test

All 9 patients with acute ON and 5/31 (16%) with past ON showed an RAPD on bedside SLFT. In contrast, none of the MS ON- patients and none of the HC had an RAPD on bedside SLFT.

## Comparative analysis of pupil diameter as assessed by ultrasound pupillometry and infrared video pupillometry

All participants tolerated UP and IVP well and none of the UP examinations had to be terminated prematurely. In one HC, IVP could not be performed due to repetitive blinking. Results of PLR assessment by IVP and UP in ON+ eyes and HC eyes are shown in Table 2.

PD at rest measured by UP and IVP correlated well in ON+ eyes ($r = 0.56$, 95% CI: 0.35; 0.72, Fig 1a) and in HC eyes ($r = 0.60$, 95% CI: 0.47; 0.72, Fig 1b). Similarly, PD after direct $L_{stim}$ correlated in both methods (ON+ eyes: $r = 0.64$, 95% CI: 0.47; 0.77, Fig 2a and HC eyes: $r = 0.63$, 95% CI: 0.50; 0.74, Fig 2b). However, the PD at rest was generally smaller in UP than in IVP measurements (difference, mean (SD), ON+ eyes: 0.44 (0.87) mm, HC eyes: 0.69 (0.80) mm, Table 2).

The constriction range measured by UP and IVP correlated moderately in ON+ eyes ($r = 0.49$, 95% CI: 0.27; 0.66, Fig 3a) and weakly in HC eyes ($r = 0.34$, 95% CI: 0.13; 0.52, Fig 3b).

Table 2. Infrared video pupillometry vs. ultrasound pupillometry in ON+ and HC eyes.

| Method | ON+ eyes (n = 52, 40 individuals) | | | HC eyes (n = 100, 50 individuals) | | |
|---|---|---|---|---|---|---|
| | IVP (n = 52) | UP (n = 52) | IVP-UP (n = 52) | IVP (n = 98) | UP (n = 100) | IVP-UP (n = 98) |
| PD at rest, mm, mean (SD) | 5.23 (1.11) | 4.80 (0.78) | 0.44 (0.87) | 5.55 (1.05) | 4.86 (0.85) | 0.69 (0.80) |
| Correlation of IVP and UP[A], $r$ (95% CI), $p$ | 0.56 (0.35; 0.72) $p = 0.001$ | | | 0.60 (0.47; 0.72) $p = 0.001$ | | |
| PD after direct $L_{stim}$, mm, mean (SD) | 3.33 (0.70) | 3.03 (0.70) | 0.30 (0.59) | 3.27 (0.67) | 3.01 (0.65) | 0.26 (0.52) |
| Correlation of IVP and UP[A], $r$ (95% CI), $p$ | 0.64 (0.47; 0.77) $p = 0.001$ | | | 0.63 (0.50; 0.74) $p = 0.001$ | | |
| Constriction range, %, mean (SD) | 36.0 (6.1) | 36.9 (10.4) | -0.9 (9.4) | 41.2 (4.0) | 38.0 (6.9) | 3.2 (6.4) |
| Correlation[A] of IVP and UP, $r$ (95% CI), $p$ | 0.49 (0.27; 0.66) $p = 0.001$ | | | 0.34 (0.13; 0.52) $p = 0.001$ | | |

Correlation[A] = correlation coefficient calculated from marginal $r^2$ values of linear mixed models accounting for repeated measures, HC = healthy controls, IVP = infrared video pupillometry, $L_{stim}$ = light stimulus, mm = millimeters, ON+ eyes = eyes with optic neuritis, PD = pupil diameter, SD = standard deviation, UP = ultrasound pupillometry, 95% CI = 95% confidence interval.

To visualize individual differences in PD measurements by UP and IVP, we analyzed PD after a direct $L_{stim}$ for all ON- eyes (n = 188) using a Bland-Altman plot (Fig 4). We found a systematic shift with smaller PD after direct $L_{stim}$ in UP as compared to IVP measurements (difference, mean (SD) 0.27 (0.56) mm) and a remarkable dispersion of individual results (95% CI: -0.83; 1.37 mm) in all groups. The relationship between PD measurements by IVP and UP was approximately linear (locally weighted scatterplot close to the mean difference in Fig 4).

## Analysis of the constriction ratio by ultrasound pupillometry

UP identified pathologic CRs (CR>1.3) in 21/121 (17%) participants, including all 9 patients with acute ON, 7/31 (23%) patients with past ON (of whom 5/7 also had an RAPD in bedside SFLT), and 5/31 (16%) MS ON- patients (none of whom had an RAPD in bedside SFLT). The CRs of HC (n = 50) were all below the threshold of CR = 1.3 (Fig 5). Since an RAPD is a sign of unilateral ON, patients with past ON were divided into two groups: those with unilateral or bilateral ON with an unequal number of left and right ON episodes (n = 26) and those with bilateral ON with an equal number of ON episodes in both eyes (n = 5). Of note, all HC were also below CR = 1.2 (Fig 5).

To confirm that MS ON- patients with a CR>1.3 indeed had no history of ON, we re-evaluated the patients' medical history and medical records. This revealed no evidence of previous ON in any of the 5 patients. In a detailed analysis of the MS ON- patients with CR>1.3, all 5 patients had pathologic CR in the right eye. Only 1/5 (20%) had a worse VA in the right eye compared to the left eye (0.63 vs. 1.25). However, in 4/5 (80%) patients the right absolute pRNFLT was mildly reduced with IEAD≥5 μm and IEPD≥5%. Taken together, these findings suggest a history of subclinical unilateral ON (Table 3).

Analysis of CRs by ROC curves showed high discriminatory power to distinguish between eyes with acute ON (n = 9) and HC eyes (n = 100, AUC = 1, 95% CI: 1; 1). However, CRs showed low discriminatory power to differentiate between HC eyes (n = 100) and eyes with past ON (n = 43, AUC = 0.52, 95% CI: 0.37; 0.68) or eyes of MS ON- patients (n = 62, AUC = 0.45, 95% CI: 0.33; 0.58).

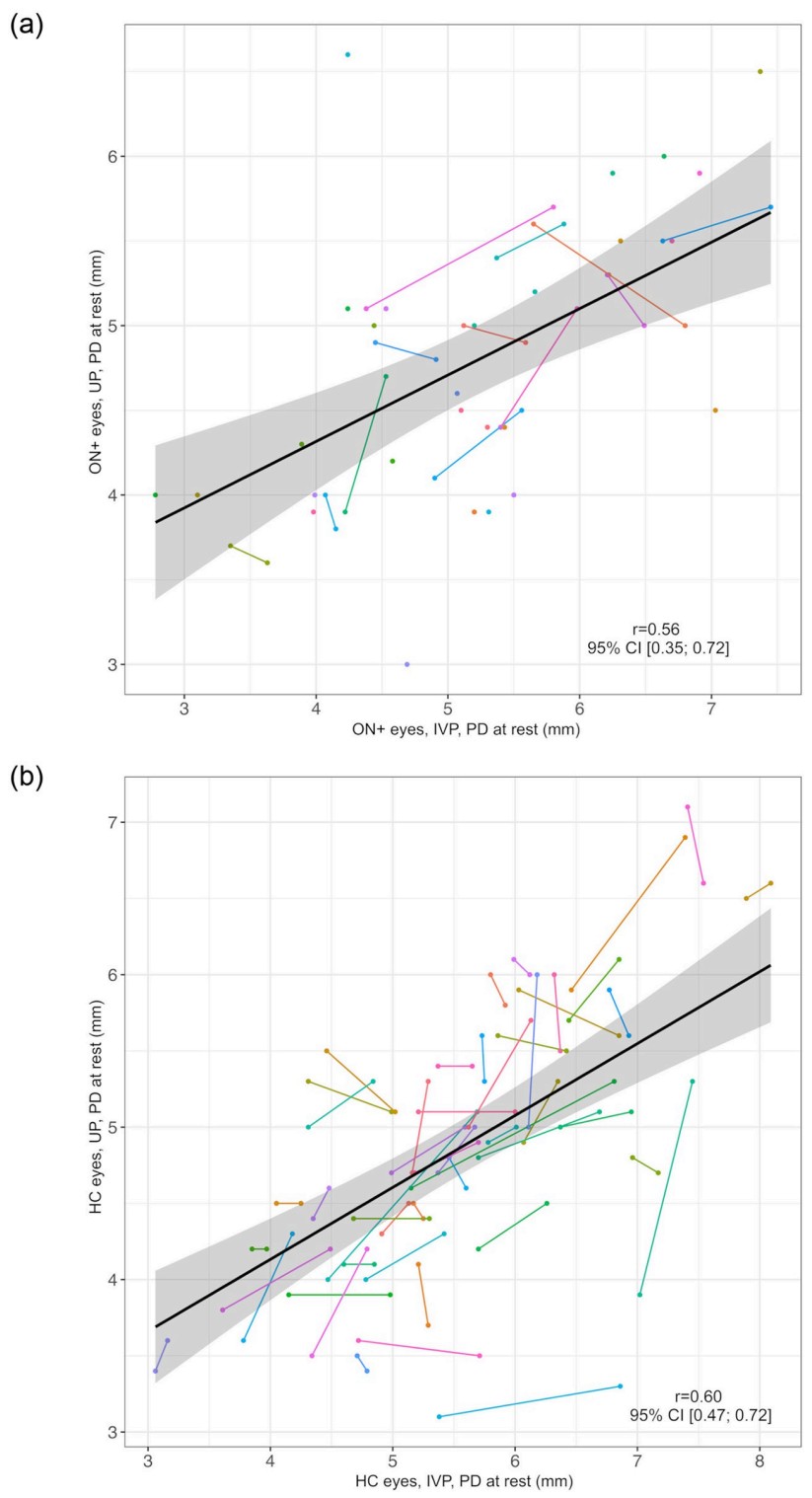

**Fig 1. Correlation of pupil diameter at rest as measured by UP and IVP.** a) ON+ eyes Black line = regression line, grey area = 95% confidence interval, IVP = infrared video pupillometry, ON+ eyes = eyes with optic neuritis, PD = pupil diameter in millimeters, r = correlation coefficient of PD at rest in IVP and UP, UP = ultrasound pupillometry, 95% CI = 95% confidence interval. Individuals are represented by different colors; two dots of the same color represent both ON+ eyes (right and left) of one individual. b) HC eyes Black line = regression line, grey area = 95% confidence interval, HC = healthy controls, IVP = infrared video pupillometry, PD = pupil diameter in

millimeters, r = correlation coefficient of PD at rest in IVP and UP, UP = ultrasound pupillometry, 95% CI = 95% confidence interval. Individuals are represented by different colors; two dots of the same color represent both ON + eyes (right and left) of one individual.

### Comparison of pupil constriction time in eyes with ON and eyes of HC

In UP, PCT was longer in eyes with acute (n = 9, mean (SD): 1020 (77) ms) and past ON (n = 43, 879 (68) ms) compared to HC eyes (n = 100, 831 (51) ms, age-adjusted mean difference (95% CI): acute ON: 195 (158; 233) ms, $p<0.001$, past ON: 46 (25; 68) ms, $p<0.001$).

### Correlation of PLR assessments with visual endpoints

VA and absolute pRNFLT in ON+ and ON- eyes are shown in Tables 4 and 5. While VA was measured in all participants, OCT data were only available in 84/121 (69%) participants.

As expected, VA of eyes with acute (n = 9, mean (SD): 0.50 (0.33)) and past ON (n = 43, 0.78 (0.40)) was lower compared to HC eyes (n = 100, 1.12 (0.23), age-adjusted mean difference (95% CI): acute ON: -0.68 (-0.86; -0.51), $p<0.001$, past ON: -0.33 (-0.44; -0.22), $p<0.001$, Table 4).

As also expected, absolute pRNFLT was lower in eyes with acute (n = 8, mean (SD): 82 (18) μm) and past ON (n = 19, 81 (19) μm) compared to HC eyes (n = 78, 95 (10) μm, age-adjusted mean difference (95% CI): acute ON: -15 (-24; -6), $p = 0.001$, past ON: -12 (-20; -5), $p = 0.001$, Table 5).

Accordingly, patients with unilateral past ON or an unequal number of past ON episodes in both eyes (n = 14) had higher IEAD (mean (SD): 13 (12) μm) and IEPD (15 (13) %) than HC (n = 39, IEAD: 3 (2) μm, IEPD: 3 (2) %, mean differences (95% CI): IEAD: 10 (6; 14) μm, $p<0.001$, IEPD: 11 (7; 16) %, $p<0.001$).

As shown in Tables 4 and 5, eyes of MS ON- patients did not differ substantially from HC eyes in terms of VA ($p = 0.586$) or absolute pRNFLT ($p = 0.383$).

The results of the NEI-VFQ, assessing the vQoL, showed very strong internal consistency (Cronbach's alpha = 0.968). The vQoL scores of MS ON- patients (n = 31, mean (SD): 94 (5)) did not differ notably from HC (n = 50, 91 (8), mean difference (95% CI): 2 (-4; 8), $p = 0.688$). In contrast, patients with acute (n = 9, mean (SD): 65 (20)) and past ON (n = 31, 80 (15)) reported lower vQoL scores than HC (n = 50, 91 (8), mean difference (95% CI): acute ON: -26 (-35; -17), $p<0.001$, past ON: -11 (-17; -5), $p<0.001$).

We evaluated the association of PLR assessment by UP or IVP with VA, absolute pRNFLT, and vQoL scores. In ON+ eyes (n = 52 eyes in 40 individuals), the correlations of the constriction range with VA and vQoL were stronger for UP assessments (VA: $r = 0.66$, 95% CI: 0.50; 0.78, vQoL: $r = 0.47$, 95% CI: 0.24; 0.66) than for IVP assessments (VA: $r = 0.52$, 95% CI: 0.36; 0.67, vQoL: $r = 0.27$, 95% CI: 0.03; 0.51) as demonstrated in Figs 6 and 7.

Furthermore, VA of ON+ eyes inversely correlated with PCT (n = 52 eyes in 40 individuals, $r = -0.44$, 95% CI: -0.58; -0.29, Fig 8) and CR (eyes with unilateral ON or unequal number of left and right ON episodes, n = 35 in 35 individuals, $r = -0.67$, 95% CI: -0.81; -0.42, Fig 9) as measured by UP. Note that PCT and CR could not be obtained with the NPi-200 pupillometer.

We found no substantial correlations between absolute pRNFLT and the constriction range as measured by UP or IVP.

## Discussion

In this study, we systematically evaluated UP as a diagnostic tool for RAPD detection in patients with ON. The main findings of this work are: (i) Strong correlation between PD

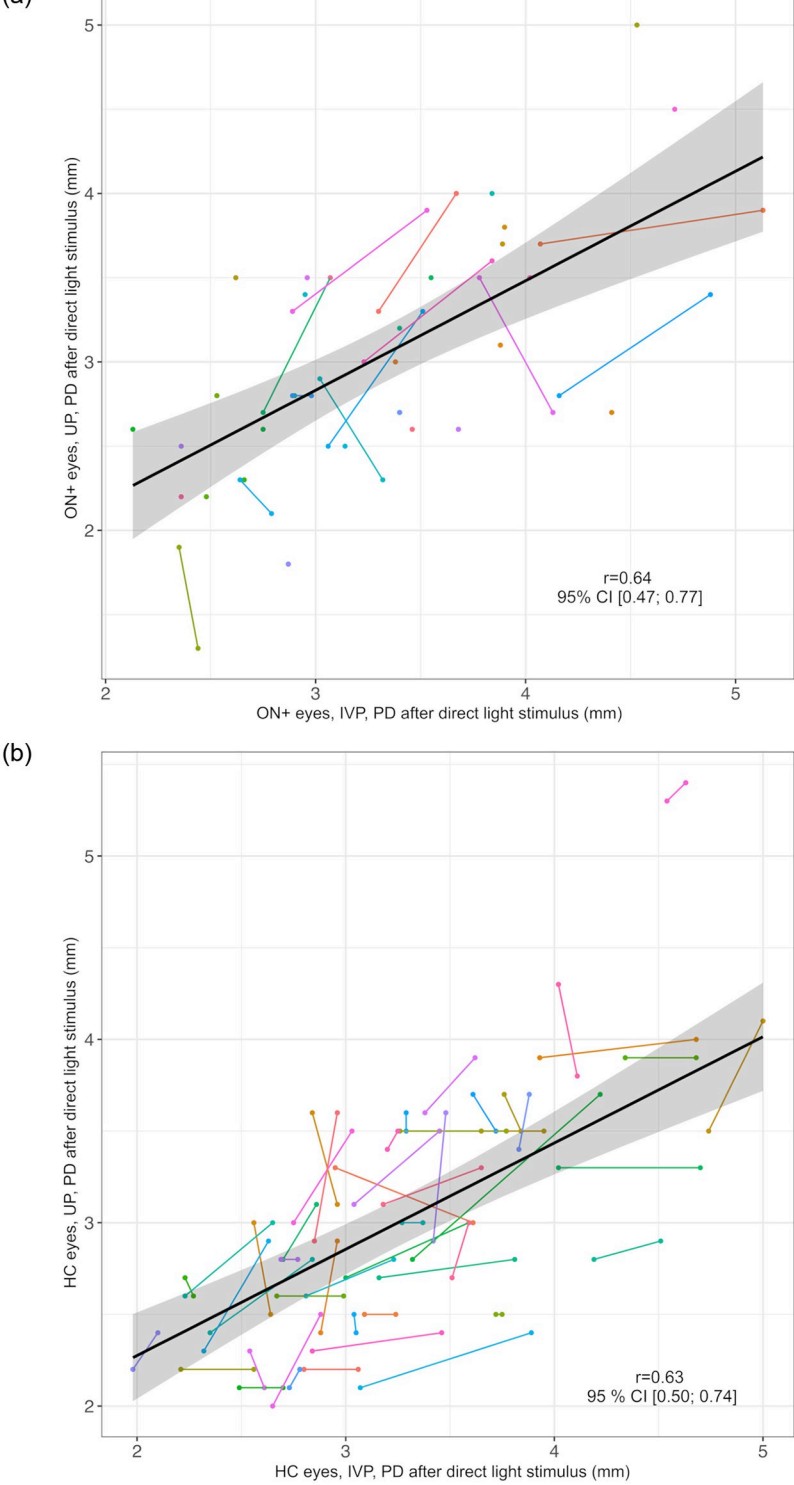

**Fig 2. Correlation of pupil diameter after direct light stimulus as measured by UP and IVP.** a) ON+ eyes Black line = regression line, grey area = 95% confidence interval, IVP = infrared video pupillometry, ON+ eyes = eyes with optic neuritis, PD = pupil diameter in millimeters, r = correlation coefficient of PD at rest in IVP and UP, UP = ultrasound pupillometry, 95%CI = 95% confidence interval. Individuals are represented by different colors; two dots of the same color represent both ON+ eyes (right and left) of one individual. b) HC eyes Black line = regression line, grey area = 95% confidence interval, HC = healthy controls, IVP = infrared video pupillometry, PD = pupil

diameter in millimeters, r = correlation coefficient of PD at rest in IVP and UP, UP = ultrasound pupillometry, 95% CI = 95% confidence interval. Individuals are represented by different colors; two dots of the same color represent both ON+ eyes (right and left) of one individual.

measurements by UP and IVP, while the overall PD was slightly smaller in UP than in IVP assessments; (ii) UP detected an RAPD in some patients with MS, who had no clinical history of ON and no RAPD as assessed by bedside SFLT; (iii) The CR as determined by UP clearly discriminated between acute ON and HC eyes; (iv) in ON+ eyes, constriction range measured by UP showed stronger correlations with VA and vQoL than constriction range measured by IVP.

The good correlation of PD measurements by UP and IVP in ON+ patients observed in the present study is consistent with previous findings in HC [14]. The systematic shift with smaller PD values in UP than in IVP measurements was most likely due to differences in the examination techniques. In fact, the eyes were closed for UP, whereas they were open for IVP. In addition, the light intensity of the penlight used in UP was much higher than that of the light diode used in IVP (70,000 vs. 1,100 lux). Finally, the scatter between UP and IVP measurements of some individual results may be due to measurement inaccuracies.

The PCT measured by UP was longer in ON+ eyes than in HC eyes, which is consistent with our previous findings [10]. Also, PCT measurements of HC were in line with the reference values (mean, (SD) left eye: 970 (261.6) ms; right eye: 967 (220) ms) for healthy subjects from our previous study [9]. PCT is a dynamic parameter to evaluate an RAPD, a clear advantage of UP over the IVP device used in this study, which could not measure PCT. Furthermore, the NPi-200 pupillometer can only illuminate and examine one eye at a time. Hence, the SFLT, which includes illumination of one eye and assessment of the consensual PLR in the contralateral eye, and which can easily be assessed by UP, could not be performed with the NPi-200 pupillometer.

All 14 patients with an RAPD as determined by bedside SFLT also had a pathologic CR as determined by UP. However, importantly, we identified 7 additional patients with pathologic CRs who did not have an RAPD as assessed by SFLT, suggesting that UP may be more sensitive in detecting an RAPD than SFLT. UP is an objective and precise measuring tool, providing consistent and reliable measurements [9], whereas the SFLT relies on subjective visual evaluation by the examiner [6, 7]. In the SFLT, results depend on the examiner's experience and judgment [8]. In contrast, UP can detect subtle changes in pupil size that might be overlooked during the SFLT. Also, UP offers the possibility to document and save results on frozen images and video sequences of the PLR, which allows for re-analysis. All these factors may contribute to the higher detection sensitivity of UP for an RAPD.

Despite no significant differences between MS ON- patients and HC in terms of VA, pRNFLT, or vQoL, 5/31 (16%) MS ON- patients had pathologic CRs as measured by UP. Of these, one patient had reduced VA and four had slightly reduced absolute pRFNLT with increased IEAD and IEPD in the affected eye [17], altogether indicating subtle optic nerve damage. In conclusion, these results suggest that UP may detect an RAPD with a higher sensitivity than the SFLT and that UP may be able to detect subclinical optic nerve damage in patients with MS.

All HC had CR<1.3, confirming the reliability of this threshold [10]. However, CR = 1.3 appears to be a conservative cut-off value, as CR = 1.2 still separated patients with acute ON from HC.

ROC curves of CRs discriminated patients with acute ON from HC with high discriminatory power. UP also discriminated patients with past ON and MS ON- patients from HC.

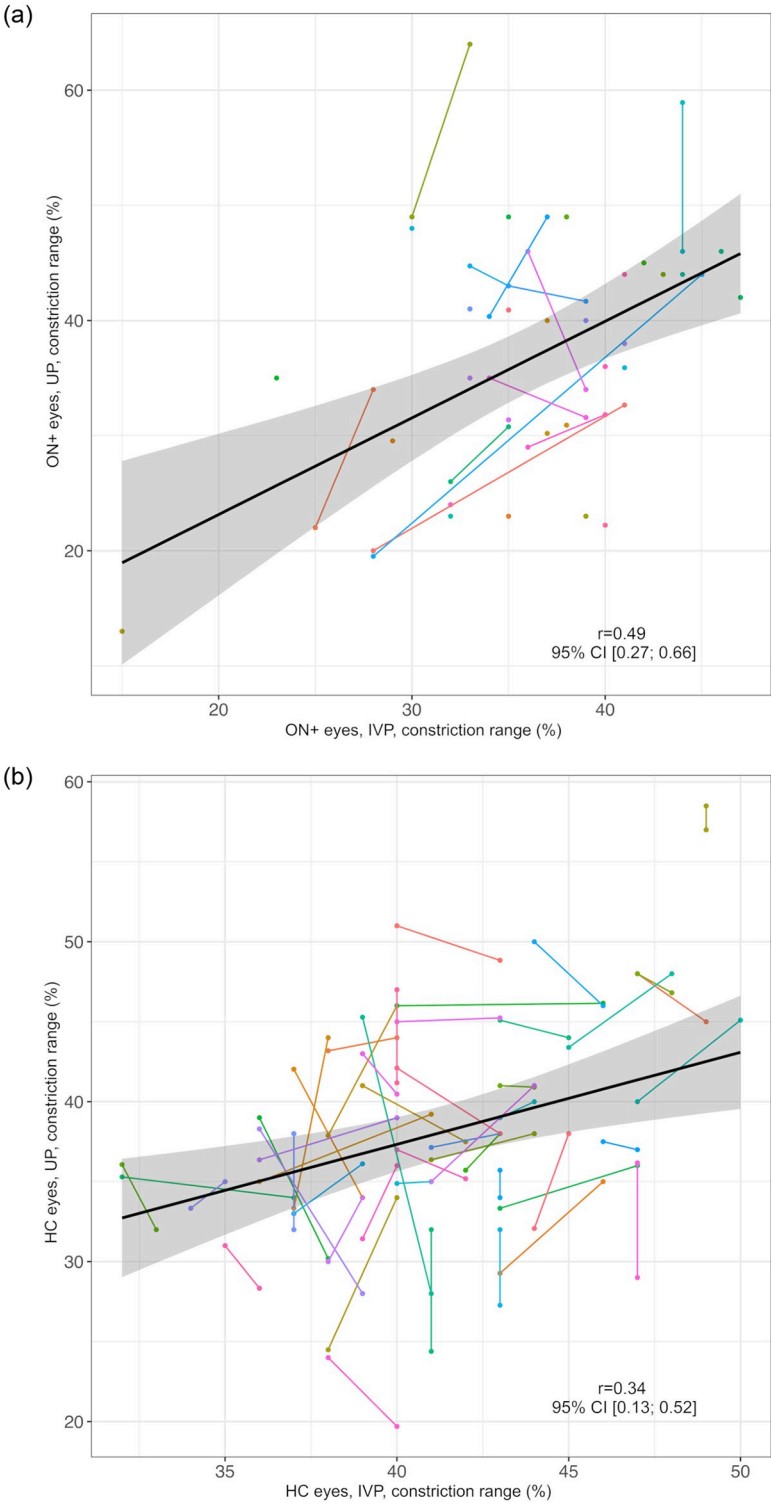

**Fig 3. Correlation of constriction range as measured by UP and IVP.** <u>a) ON+ eyes</u> Black line = regression line, grey area = 95% confidence interval, IVP = infrared video pupillometry, ON+ eyes = eyes with optic neuritis, r = correlation coefficient of constriction range in IVP and UP, UP = ultrasound pupillometry, 95% CI = 95% confidence interval. Individuals are represented by different colors; two dots of the same color represent both ON+ eyes (right and left) of one individual. <u>b) HC eyes</u> Black line = regression line, grey area = 95% confidence interval, HC = healthy controls, IVP = infrared video pupillometry, r = correlation coefficient of constriction range in IVP and UP, UP = ultrasound

pupillometry, 95% CI = 95% confidence interval. Individuals are represented by different colors; two dots of the same color represent both ON+ eyes (right and left) of one individual.

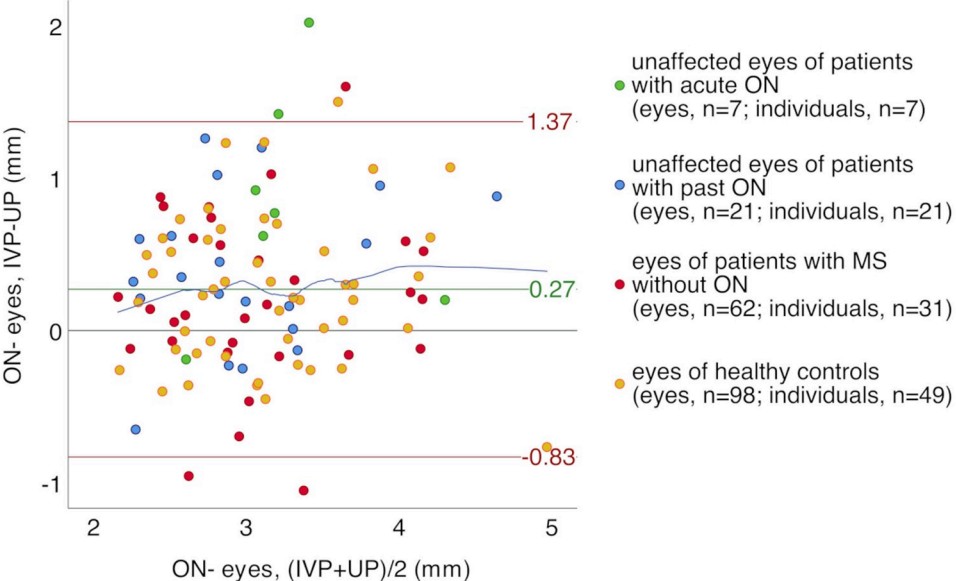

**Fig 4. Bland-Altman plot of pupil diameter as measured by UP and IVP after direct light stimulus in ON- eyes.**
Green line = mean difference, red lines = 95% confidence interval, blue line = locally weighted scatterplot smoothing, IVP = infrared video pupillometry, $L_{stim}$ = light stimulus, mm = millimeters, MS = multiple sclerosis, ON = optic neuritis, ON- eyes = eyes without optic neuritis, PD = pupil diameter, UP = ultrasound pupillometry.

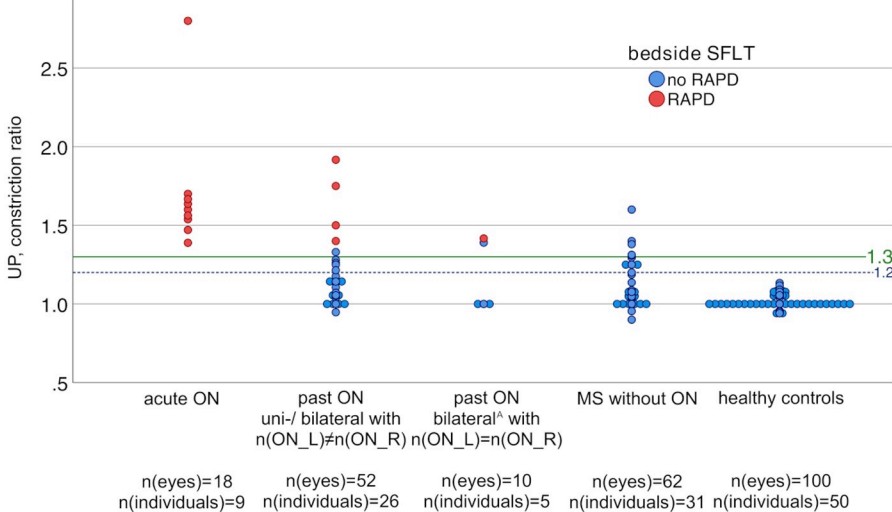

**Fig 5. Constriction ratios in UP by study groups.** Bilateral[A] = bilateral simultaneous or sequential ON episodes, MS = multiple sclerosis, ON = optic neuritis, n(ON_L/R) = number of left/right ON episodes, RAPD = relative afferent pupillary defect, SFLT = swinging flashlight test, UP = ultrasound pupillometry. The x-axis represents the study groups: patients with acute ON (n = 9), patients with past unilateral ON or bilateral ON with n(ON_L)≠n(ON_R) (n = 26), patients with past bilateral ON with n(ON_L) = n(ON_R) (n = 5), patients with MS without ON (n = 31), and healthy controls (n = 50). The y-axis represents the participant's constriction ratio in UP.

**Table 3. Demographic and clinical data of MS ON- patients with pathologic constriction ratio in UP.**

| Patient | Sex | Age (years) | MS type | EDSS | CR, right/ left | VA, right/ left | pRNFLT, right/left (μm) | IEAD (μm) | IEPD (%) |
|---|---|---|---|---|---|---|---|---|---|
| 1 | male | 32 | RRMS | 1.0 | 1.60/ 0.92 | 1.25/ 1.25 | 97/ 96 | 1 | 1 |
| 2 | female | 39 | RRMS | 2.5 | 1.31/ 0.85 | 1.25/ 1.25 | 90/ 95 | 5 | 5 |
| 3 | male | 49 | PPMS | 4.0 | 1.38/ 0.55 | 0.63/ 1.25 | 87/ 96 | 9 | 9 |
| 4 | male | 50 | RRMS | 2.0 | 1.40/ 0.71 | 1.25/ 1.25 | 89/ 94 | 5 | 5 |
| 5 | female | 62 | RRMS | 4.0 | 1.31/ 0.85 | 1.00/ 1.00 | 81/ 87 | 6 | 7 |

CR = constriction ratio, EDSS = Expanded Disability Status Scale, IEAD = intereye absolute difference, IEPD = intereye percentage difference, MS = multiple sclerosis, MS ON- = patients with multiple sclerosis without a history of ON, μm = micrometers, pRNFLT = absolute peripapillary retinal nerve fiber layer thickness, RRMS = relapsing remitting multiple sclerosis, PPMS = primary progressive multiple sclerosis, UP = ultrasound pupillometry, VA = visual acuity.

**Table 4. Visual acuity in ON+ and ON- eyes.**

| Study group | ON+ eyes (n = 52, 40 individuals) | | ON- eyes (n = 190, 111 individuals) | | | | |
|---|---|---|---|---|---|---|---|
| | acute ON (n = 9) | past ON (n = 43) | unaffected eyes acute ON (n = 7) | unaffected eyes past ON (n = 21) | MS ON- (n = 62) | HC (n = 100) |
| VA, mean (SD) | 0.50 (0.33) | 0.78 (0.40) | 1.21 (0.09) | 1.06 (0.24) | 1.12 (0.20) | 1.12 (0.23) |
| Difference vs. HC (95% CI) | -0.68 (-0.86; -0.51) | -0.33 (-0.44; -0.22) | -0.03 (-0.23; 0.16) | -0.09 (-0.21; 0.03) | -0.03 (-0.14; 0.08) | n.a. |
| $p$-value, vs. HC[A] | <0.001 | <0.001 | 0.725 | 0.138 | 0.586 | n.a. |

Difference vs. HC = model based mean difference comparing each group to HC adjusted for age, HC = healthy controls, MS ON- = patients with multiple sclerosis without a history of ON, n.a. = not applicable, ON+/- eyes = eyes with or without optic neuritis, $p$-value vs. HC[A] = $p$-values based on age-adjusted linear mixed model for comparing each group with HC, SD = standard deviation, VA = visual acuity, 95% CI = 95% confidence interval.

**Table 5. Absolute peripapillary retinal nerve fiber layer thickness in ON+ and ON- eyes.**

| Study group | ON+ eyes (n = 27, 22 individuals) | | ON- eyes (n = 141, 79 individuals) | | | |
|---|---|---|---|---|---|---|
| | acute ON (n = 8) | past ON (n = 19) | unaffected eyes acute ON (n = 6) | unaffected eyes past ON (n = 11) | MS ON- (n = 46) | HC (n = 78) |
| pRNFLT, μm, mean (SD) | 82 (18) | 81 (19) | 97 (6) | 93 (21) | 93 (11) | 95 (10) |
| Difference vs. HC (95% CI) | -15 (-24; -6) | -12 (-20; -5) | -7 (-17; 2) | -2 (-10; 6) | -3 (-9; 4) | n.a. |
| $p$-value, vs. HC[A] | 0.001 | 0.001 | 0.144 | 0.601 | 0.383 | n.a. |

Difference vs. HC = model based mean difference comparing each group to HC adjusted for age, HC = healthy controls, MS ON- = patients with multiple sclerosis without a history of ON, μm = micrometers, n.a. = not applicable, ON+/- eyes = eyes with or without optic neuritis, pRNFLT = peripapillary retinal nerve fiber layer thickness, $p$-value vs. HC[A] = $p$-values based on age-adjusted linear mixed model for comparing each group with HC, SD = standard deviation, 95% CI = 95% confidence interval.

Thus, the CR method has a high diagnostic potential to separate pathologic eyes from healthy eyes.

As expected, ON+ patients had reduced VA, pRNFLT, and vQoL compared to HC [21, 26]. Of note, acute ON typically causes a swelling of the pRNFLT [27]. Here, patients with acute ON had reduced absolute pRNFLT. We attribute this to previous episodes of ON and the time elapsed between symptom onset and the examination. Importantly, correlations of VA and

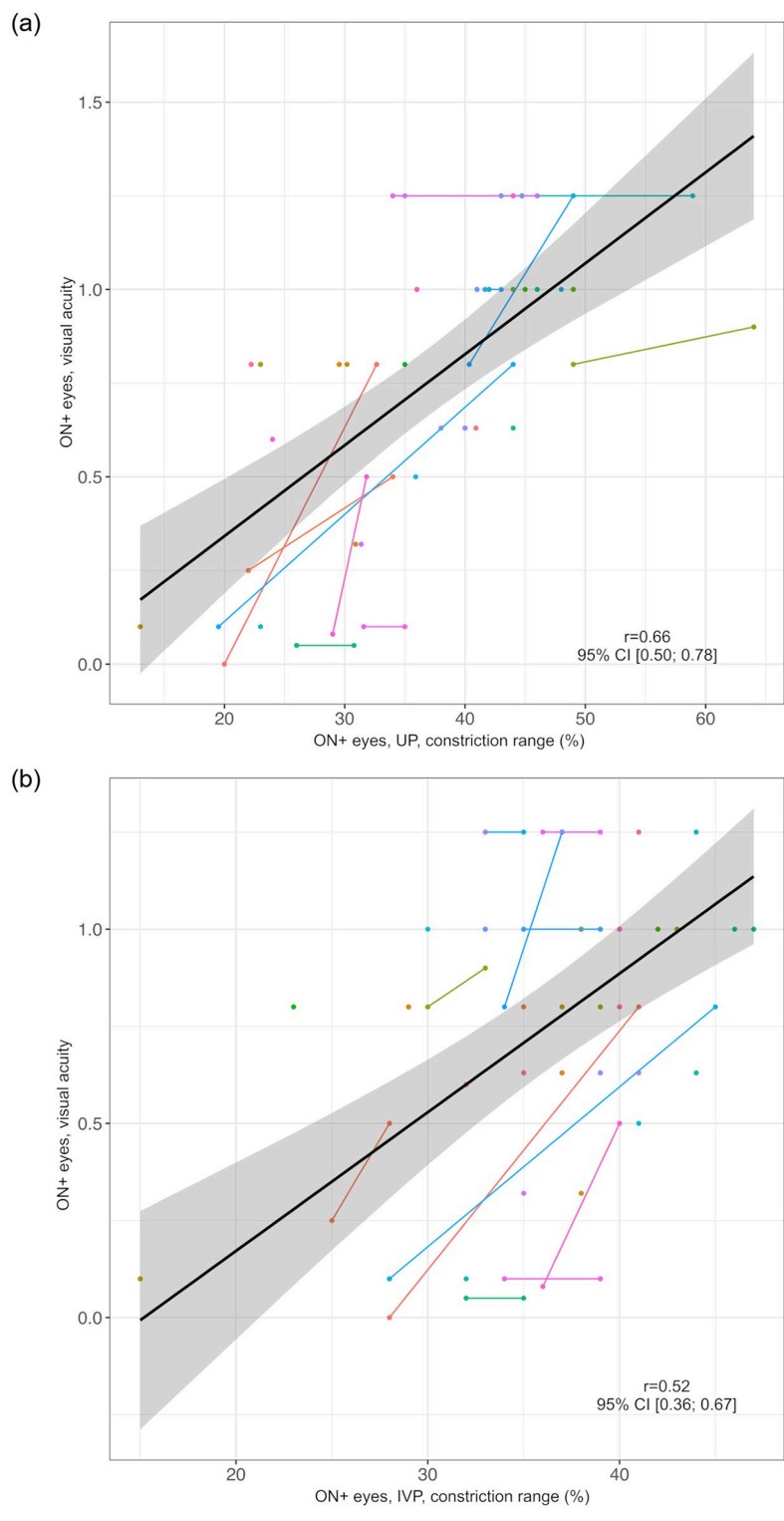

**Fig 6. Correlation of constriction range as measured by UP and IVP with visual acuity in ON+ eyes.** a) UP Black line = regression line, grey area = 95% confidence interval, ON+ eyes = eyes with optic neuritis, r = correlation coefficient of constriction range in UP and visual acuity, UP = ultrasound pupillometry, 95% CI = 95% confidence interval. Individuals are represented by different colors; two dots of the same color represent both ON+ eyes (right and left) of one individual. b) IVP Black line = regression line, grey area = 95% confidence interval, IVP = infrared video pupillometry, ON+ eyes = eyes with optic neuritis, r = correlation coefficient of constriction range in IVP and visual

acuity, 95% CI = 95% confidence interval. Individuals are represented by different colors; two dots of the same color represent both ON+ eyes (right and left) of one individual.

vQoL in ON+ eyes were stronger with UP than with IVP. Additionally, PCT and CR as measured by UP correlated well with VA in ON+ eyes.

Altogether, our results suggest that UP can detect an RAPD with similar performance characteristics as IVP. We consider the wider availability of this easy to perform diagnostic method and the possibility to perform UP in cases of obstructed eyelid retraction [28] as advantages of UP over IVP.

Among the strengths of this study is the inclusion of various objective (VA, OCT) and subjective (VQoL) visual outcome parameters for ON. VQoL is a subjective measure that captures the patient's perspective on the impact of visual impairments on their daily life and overall well-being. The correlation of UP and IVP parameters with VQoL therefore indicates that these parameters may reflect the real-world impact of ON on a patient's daily life. Changes in the pRNFLT, as measured by OCT, indicates structural optic nerve damage. Altogether, our multifaceted approach enabled cross-validation of UP and IVP findings with other established objective and subjective outcome measures of ON, enhancing the clinical relevance of our findings.

A potential limitation of this study is observer bias, as all examinations and interpretations were performed by the same examiner (FrS). Furthermore, inclusion of patients with bilateral ON may have made the detection of an RAPD more difficult. Another possible limitation of our study is that we could not systematically control for potential effects of stimulants, such as caffeine or nicotine, on the pupillary reaction. However, we consider it unlikely that any possible use of stimulants might have significantly affected the results of our study, as the RAPD detects relative differences of the pupillary reaction of both eyes. While stimulants could possibly affect pupillary reactions globally, they appear unlikely to affect relative differences of pupillary reactions of both eyes. Finally, the analysis of pRNFLT in patients with acute ON may have limited interpretability.

To further evaluate the diagnostic accuracy of UP, blinded case-control studies with larger numbers of cases would be of interest. Future studies could also utilize recently emerging wireless portable ultrasound and IVP devices capable of assessing consensual $L_{stim}$ and PCT to facilitate comparison between both methods and to determine in which setting IVP or UP may be preferable to assess an RAPD.

## Conclusions

This study shows that UP is a suitable method for objective detection and quantification of an RAPD with similar performance characteristics as IVP. RAPD detection by UP is more accurate and sensitive than clinical SFLT examination with a penlight and may therefore be able to detect subclinical optic nerve damage in patients with MS. Our results suggest that UP is a valuable diagnostic tool, which can be used in clinical routine for objective detection of an RAPD. We propose a rapid and simplified standardized protocol for RAPD detection by UP to be used in clinical routine for the detection of unilateral ON, but also other optic neuropathies.

## Clinical examination protocol for UP

1. Place a linear ultrasound probe on the closed lower left eyelid, focusing on the pupil.

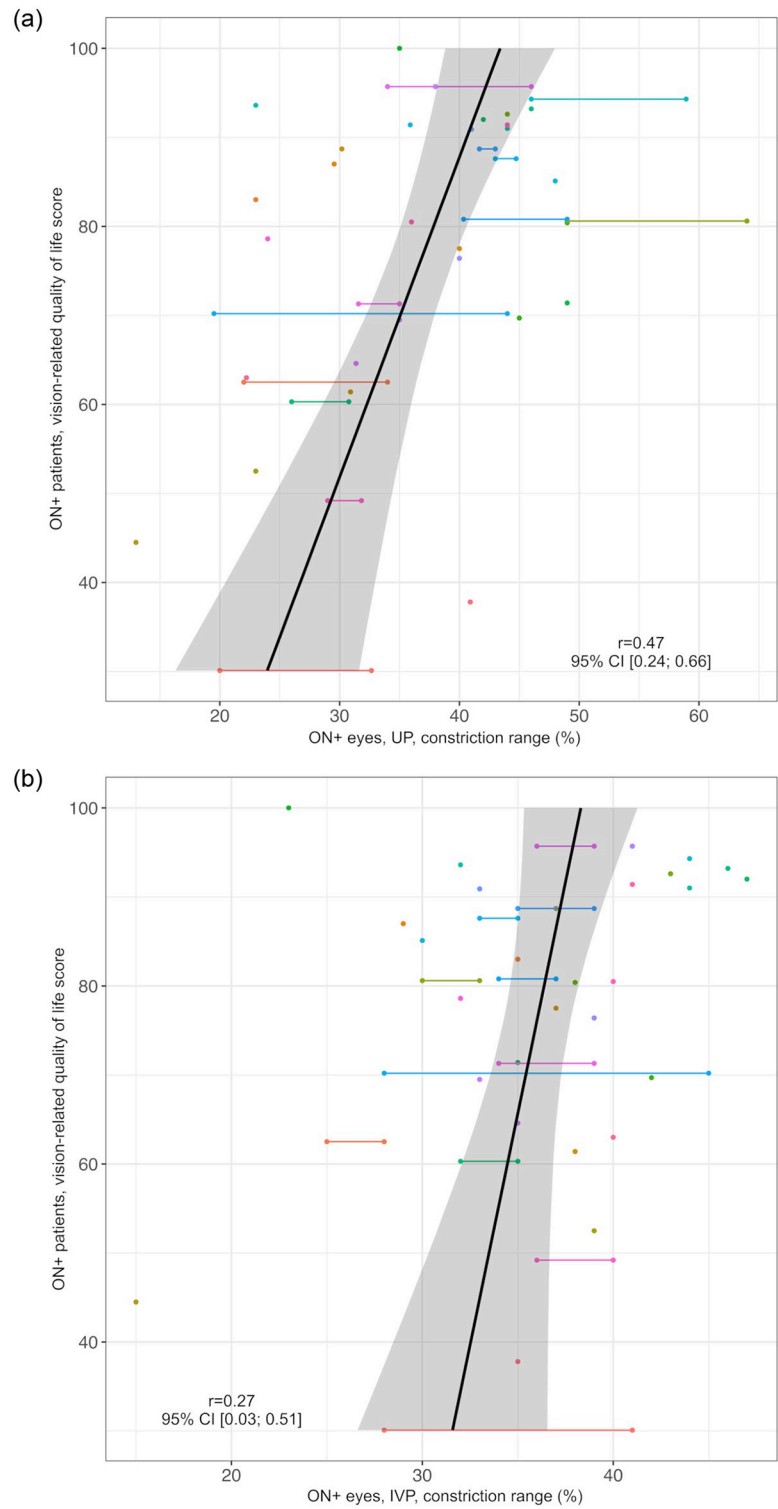

**Fig 7. Correlation of constriction range as measured by UP and IVP in ON+ eyes with vision-related quality of life scores.** a) UP Black line = regression line, grey area = 95% confidence interval, ON+ eyes/ patients = eyes/ patients with optic neuritis, r = correlation coefficient of constriction range in UP and vision-related quality of life score, UP = ultrasound pupillometry, 95% CI = 95% confidence interval. Individuals are represented by different colors; two dots of the same color represent both ON+ eyes (right and left) of one individual. b) IVP Black line = regression line, grey area = 95% confidence interval, IVP = infrared video pupillometry, ON+ eyes/ patients = eyes/ patients with optic

neuritis r = correlation coefficient of constriction range in IVP and vision-related quality of life score, 95% CI = 95% confidence interval. Individuals are represented by different colors; two dots of the same color represent both ON + eyes (right and left) of one individual.

2. Perform the SFLT with a penlight illuminating first the contralateral (right) eye and then the ipsilateral (left) eye, while recording a 5-second video of the left PLR.

3. Repeat the same procedure with the probe on the right eye.

4. Measure PD at rest and PD after direct and consensual $L_{stim}$ on frozen images of the video sequences.

5. Compute the direct and consensual constriction amplitudes and determine the CR, the quotient of the consensual and direct constriction amplitude.

6. A CR>1.3 indicates an ipsilateral RAPD.

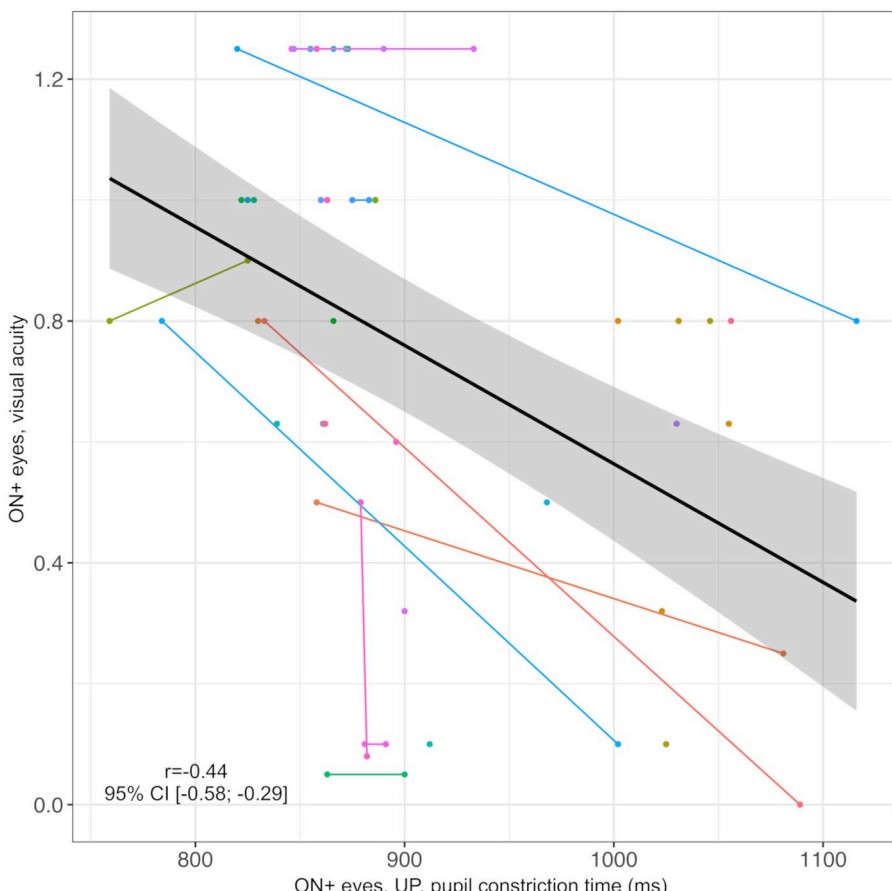

**Fig 8. Correlation of pupil constriction time as measured by UP with visual acuity in ON+ eyes.** Black line = regression line, grey area = 95% confidence interval, ms = milliseconds, ON+ eyes = eyes with optic neuritis, r = correlation coefficient of pupil constriction time in UP and visual acuity, UP = ultrasound pupillometry, 95% CI = 95% confidence interval. Individuals are represented by different colors; two dots of the same color represent both ON+ eyes (right and left) of one individual.

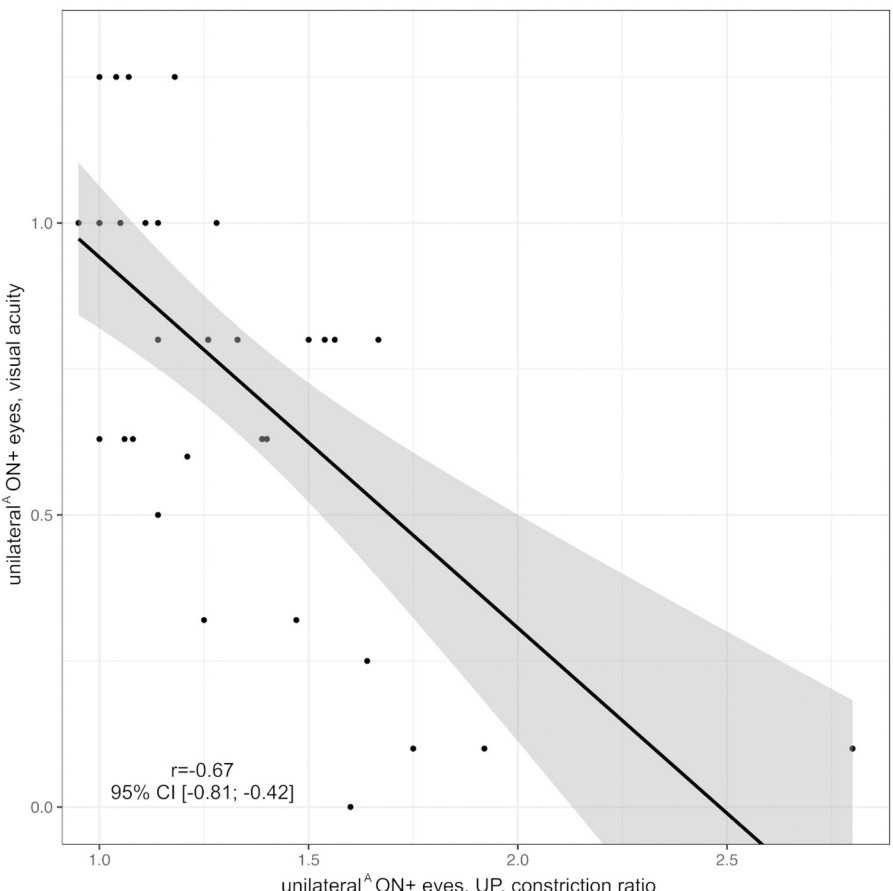

**Fig 9. Correlation of constriction ratio as measured by UP with visual acuity in unilateral[A] ON+ eyes.** Black line = regression line, grey area = 95% confidence interval, ON = optic neuritis, r = correlation coefficient of constriction ratio in UP and visual acuity, unilateral[A] ON+ eyes = eyes with unilateral ON or bilateral ON episodes with an unequal number of ON episodes in the left and right eye, UP = ultrasound pupillometry, 95% CI = 95% confidence interval.

## Supporting information

**S1 Table. Data set.** CIS = clinically isolated syndrome, EDSS = Expanded Disability Status Scale, HC = healthy controls, IVP = infrared video pupillometry, mm = millimeters, MS = multiple sclerosis, ms = milliseconds, µm = micrometers, NMOSD = neuromyelitis optica spectrum disorder, OCT = optical coherence tomography, ON = optic neuritis, PCT = pupil constriction time, PD = pupil diameter, PPMS = primary progressive multiple sclerosis, pRNFLT = absolute peripapillary retinal nerve fiber layer thickness, RAPD = relative afferent pupillary defect, RRMS = relapsing remitting multiple sclerosis, SPMS = secondary progressive multiple sclerosis, UP = ultrasound pupillometry.
(XLSX)

## Author Contributions

**Conceptualization:** Franziska Siebald, Klemens Ruprecht, Felix A. Schmidt.

**Data curation:** Franziska Siebald.

**Formal analysis:** Franziska Siebald, Ulrike Grittner.

**Funding acquisition:** Klemens Ruprecht.

**Investigation:** Franziska Siebald.

**Methodology:** Franziska Siebald, Charlotte Bereuter, Klemens Ruprecht, Felix A. Schmidt.

**Project administration:** Klemens Ruprecht, Felix A. Schmidt.

**Resources:** Ulrike Grittner, Carolin Otto, Charlotte Bereuter, Jan Klonner, Klemens Ruprecht, Felix A. Schmidt.

**Software:** Ulrike Grittner.

**Supervision:** Klemens Ruprecht, Felix A. Schmidt.

**Validation:** Franziska Siebald, Ulrike Grittner, Klemens Ruprecht, Felix A. Schmidt.

**Visualization:** Franziska Siebald, Ulrike Grittner.

**Writing – original draft:** Franziska Siebald.

**Writing – review & editing:** Franziska Siebald, Ulrike Grittner, Carolin Otto, Charlotte Bereuter, Hanna G. Zimmermann, Lutz Harms, Jan Klonner, Stephan J. Schreiber, Friedemann Paul, Klemens Ruprecht, Felix A. Schmidt.

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
