## [Decision Letter · Decision Letter 0]

30 Apr 2024

PONE-D-24-06833Ultrasound pupillometry for the detection of a relative afferent pupillary defect (RAPD): systematic evaluation in patients with optic neuritis and comparison with infrared video pupillometryPLOS ONE

Dear Dr. Schmidt,

Thank you for submitting your manuscript to PLOS ONE. After careful consideration, we feel that it has merit but does not fully meet PLOS ONE’s publication criteria as it currently stands. Therefore, we invite you to submit a revised version of the manuscript that addresses the points raised during the review process.

We look forward to receiving your revised manuscript.

Kind regards,

Manabu Sakakibara, Ph.D.

Academic Editor

PLOS ONE

Journal Requirements:

"I have read the journal's policy and the authors of this manuscript have the following competing interests: Felix A. Schmidt received support from the Berlin Institute of Health (BIH) and the Stiftung Charité, Berlin, Germany (Clinician Scientist Grant). Klemens Ruprecht received research support from Novartis, Basel, Switzerland, Merck Serono, German Ministry of Education and Research, Stiftung Charité (BIH Clinical Fellow Program), Germany, European Union (821283-2), Guthy Jackson Charitable Foundation, Beverly Hills, CA, USA, and Arthur Arnstein Foundation, Berlin, Germany; received speaker honoraria from Novartis and Virion Serion, Würzburg, Germany, and travel grants from Guthy Jackson Charitable Foundation. 

All other authors have declared that no competing interests exist."

**Additional Editor Comments:**

Two experts in the field have carefully reviewed the manuscript entitled "Ultrasound pupillometry for the detection of a relative afferent pupillary defect (RAPD): systematic evaluation in patients with optic neuritis and comparison with infrared video pupillometry". Their comments are appended below.

The first referee raised several serious concerns which should be considered before publication, especially Introduction section for more easy understanding.

The second reviewer pointed out that there are many mistakes in the manuscript.

Reviewers' comments:

Reviewer's Responses to Questions

**Comments to the Author**

1. Is the manuscript technically sound, and do the data support the conclusions?

Reviewer #1: Yes

Reviewer #2: Yes

2. Has the statistical analysis been performed appropriately and rigorously? 

Reviewer #1: Yes

Reviewer #2: Yes

3. Have the authors made all data underlying the findings in their manuscript fully available?

Reviewer #1: Yes

Reviewer #2: Yes

4. Is the manuscript presented in an intelligible fashion and written in standard English?

Reviewer #1: Yes

Reviewer #2: No

5. Review Comments to the Author

Reviewer #1: It has been a pleasure to review the authors’ manuscript. In this exploratory experiment, the authors evaluated the optic neuritis (ON) diagnostic utility of B-mode ultrasound pupillometry (UP) against infrared video pupillometry (IVP) across multiple visual endpoints of ON. The authors concluded that UP performed similarly to IVP and RAPD assessment by UP may detect subclinical ON damage in patients with multiple sclerosis. They also offered a standardized protocol for RAPD detection by UP in routine clinical evaluation. Multiple merits are present in the manuscript including detailed descriptions of the methods and report of the results. However, several areas of improvement need to be addressed until I endorse the publication.

Major points:

- The first paragraph in the introduction section has a strong flow. To aid in readers’ understanding, could the authors include a brief review of how the gold standard IVP has been applied to detect ON? Additional background information on the technical differences and general pros and cons in the clinical settings between IVP and UP in measuring PLR would be valuable (even though some information was discussed in the discussion section). Please also highlight the benefits of discussing other visual endpoints of ON (including both objective and subjective measures).

- Line 475: Could the authors elaborate on the possible factors in the higher detection sensitivity of UP?

- In addition to the quality plots, as part of good practice in pupillometry studies, could the authors also provide the aggregated waveform of pupillary changes during the tests?

Minor points:

- Line 81-82: there appears to be only one sentence in a paragraph.

- Line 85 (citation 12): please provide the correlation value of the strong correlation as a reference for the effect size in the present study.

- Line 115: did the authors also control for the use of stimulants as caffeine in the sample?

- Line 155: please provide one sample item from the questionnaire and report the likert scale as well as its psychometric properties based on literature. Please also state internal reliability as Cronbach’s alpha.

- Line 190-191: please discuss the exploratory nature of this study in the introduction.

- Could the authors also provide reference values of the key ON related measures (e.g., PCT) from much larger datasets for readers to compare with the sample in this study?

Reviewer #2: 1.The author mentioned the data in tables and figure in result section to support the conclusion under discussion section of manuscript.

2.The explained this in 'Statistics' section of 'Materials and methods' in manuscript.

3.The author mentioned it in "Data Availability Statement in the manuscript PDF file".

4.There are many mistakes in subheading and text of manuscript.

6. PLOS authors have the option to publish the peer review history of their article (what does this mean?). If published, this will include your full peer review and any attached files.

Reviewer #1: **Yes: **Xi Yang

Reviewer #2: No

---

## [Author Response · Author response to Decision Letter 0]

13 Jul 2024

Response to Journal Requirements:

Response:

We confirm that our manuscript meets PLOS ONE's style requirements, including those for file naming.

"I have read the journal's policy and the authors of this manuscript have the following competing interests: Felix A. Schmidt received support from the Berlin Institute of Health (BIH) and the Stiftung Charité, Berlin, Germany (Clinician Scientist Grant). Klemens Ruprecht received research support from Novartis, Basel, Switzerland, Merck Serono, German Ministry of Education and Research, Stiftung Charité (BIH Clinical Fellow Program), Germany, European Union (821283-2), Guthy Jackson Charitable Foundation, Beverly Hills, CA, USA, and Arthur Arnstein Foundation, Berlin, Germany; received speaker honoraria from Novartis and Virion Serion, Würzburg, Germany, and travel grants from Guthy Jackson Charitable Foundation. 

All other authors have declared that no competing interests exist."

Please confirm that this does not alter your adherence to all PLOS ONE policies on sharing data and materials, by including the following statement: ""This does not alter our adherence to PLOS ONE policies on sharing data and materials.” 

If there are restrictions on sharing of data and/or materials, please state these. Please note that we cannot proceed with consideration of your article until this information has been declared. 

Response:

We updated the Competing Interests as follows:

"I have read the journal's policy and the authors of this manuscript have the following competing interests: Felix A. Schmidt received support from the Berlin Institute of Health (BIH) and the Stiftung Charité, Berlin, Germany (Clinician Scientist Grant). Klemens Ruprecht received research support from Novartis, Basel, Switzerland, Merck Serono, German Ministry of Education and Research, Stiftung Charité (BIH Clinical Fellow Program), Germany, European Union (821283-2), Guthy Jackson Charitable Foundation, Beverly Hills, CA, USA, and Arthur Arnstein Foundation, Berlin, Germany; received speaker honoraria from Novartis and Virion Serion, Würzburg, Germany, and travel grants from Guthy Jackson Charitable Foundation. 

All other authors have declared that no competing interests exist."

"This does not alter our adherence to PLOS ONE policies on sharing data and materials.”

3. Please include captions for your Supporting Information files at the end of your manuscript, and update any in-text citations to match accordingly.

Response:

We have included captions for the Supporting Information files and updated in-text citations.

Response to Reviewers

Reviewer #1:

Reviewer’s point:

It has been a pleasure to review the authors’ manuscript. In this exploratory experiment, the authors evaluated the optic neuritis (ON) diagnostic utility of B-mode ultrasound pupillometry (UP) against infrared video pupillometry (IVP) across multiple visual endpoints of ON. The authors concluded that UP performed similarly to IVP and RAPD assessment by UP may detect subclinical ON damage in patients with multiple sclerosis. They also offered a standardized protocol for RAPD detection by UP in routine clinical evaluation. Multiple merits are present in the manuscript including an interesting experimental design, a coherent introduction section, detailed descriptions of the methods and report of the results, as well as a thorough discussion of the limitations.

Response: 

We would like to thank the Reviewer for this comment.

Reviewer’s point:

The first paragraph in the introduction section has a strong flow. To aid in readers’ understanding, could the authors include a brief review of how the gold standard IVP has been applied to detect ON?

Response: 

We appreciate this thoughtful feedback. In response to this point, we added the following sentences as well as two additional references, i.e. Blazcek et al. (2012) and Cohen et al. (2015), to the introduction section:

"Infrared video pupillometry (IVP), the current gold standard for objective assessment of the PLR, provides precise quantitative measurements and allows for the analysis of the PLR through high-resolution video imaging [11]. For IVP, the infrared video camera records the PLR of the ipsilateral open eye after an Lstim of the integrated light diode with a fixed distance between the eye and the light source. IVP has previously been applied for the detection of an RAPD in patients with ON where it was shown to detect an RAPD with high sensitivity and specificity [12, 13]." (pages 5-6, lines 86-95)

Reviewer’s point:

Additional background information on the technical differences and general pros and cons in the clinical settings between IVP and UP in measuring PLR would be valuable (even though some information was discussed in the discussion section).

Response:

We appreciate this suggestion. To provide more background information on the technical differences and general pros and cons in the clinical settings between IVP and UP in measuring PLR we have added the following sentence to the introduction:

"In UP, patients are examined with their eyes closed to prevent contact of the ultrasound gel with the eye, while the examiner uses an external manual light source (penlight)." (page 5, lines 82-84)

Furthermore, we added the following passage:

"However, unlike ultrasound, IVP devices are rarely available in non-specialized clinical settings. Furthermore, IVP assessment requires the ability to fixate on a target and to keep the eyes open throughout the examination, whereas UP does not require patient cooperation." (page 6, lines 95-98)

Reviewer’s point:

Please also highlight the benefits of discussing other visual endpoints of ON (including both objective and subjective measures).

Response:

We thank the Reviewer for this comment. In response, we added the following passage to the discussion section:

"Among the strengths of this study is the inclusion of various objective (VA, OCT) and subjective (VQoL) visual outcome parameters for ON. VQoL is a subjective measure that captures the patient's perspective on the impact of visual impairments on their daily life and overall well-being. The correlation of UP and IVP parameters with VQoL therefore indicates that these parameters may reflect the real-world impact of ON on a patient’s daily life. Changes in the pRNFLT, as measured by OCT, indicates structural optic nerve damage. Altogether, our multifaceted approach enabled cross-validation of UP and IVP findings with other established objective and subjective outcome measures of ON, enhancing the clinical relevance of our findings." (page 27, lines 568-576) 

Reviewer’s point:

Line 475: Could the authors elaborate on the possible factors in the higher detection sensitivity of UP?

Response:

Yes, we have now included the following sentences in the discussion section:

"UP is an objective and precise measuring tool, providing consistent and reliable measurements, whereas the SFLT relies on subjective visual evaluation by the examiner. In the SFLT, results depend on the examiner’s experience and judgment. In contrast, UP can detect subtle changes in pupil diameter that might be overlooked during the SFLT. Also, UP offers the possibility to document and save results on frozen images and video sequences of the PLR, which allows for re-analysis. All these factors may contribute to the higher detection sensitivity of UP for an RAPD." (page 25, lines 532-538)

Reviewer’s point:

In addition to the quality plots, as part of good practice in pupillometry studies, could the authors also provide the aggregated waveform of pupillary changes during the tests?

Response:

We concur with the Reviewer's assertion that the aggregation of pupillary changes during the tests would be interesting. However, in this study, we employed pupillometry assessments by ultrasound pupillometry (UP) and infrared video pupillometry (IVP) to measure two metric values: the maximum pupil diameter (PD) at rest and the minimum PD after a light stimulus. As a result, we did not record continuous data, which precluded analyses of the aggregated waveform of pupillary changes.

Reviewer’s point:

Line 81-82: there appears to be only one sentence in a paragraph.

Response:

We thank the Reviewer for bringing this formatting issue to our attention. To provide further clarification regarding the technical application of ultrasound pupillometry (UP), we have added another sentence to the respective paragraph.

“In UP, patients are examined with their eyes closed to prevent contact of the ultrasound gel with the eye, while the examiner uses an external manual light source (penlight).” (page 5, lines 82-84)

Reviewer’s point:

Line 85 (citation 12): please provide the correlation value of the strong correlation as a reference for the effect size in the present study.

Response: 

We agree with the Reviewer that the actual correlation coefficient is a crucial information. We now included the correlation coefficient from the mentioned study (Farina et al., 2021):

“(Pearson’s correlation coefficient, r=0.831, p < 0.01)” Page 6, lines 98-99

Reviewer’s point:

Line 115: did the authors also control for the use of stimulants as caffeine in the sample?

Response:

We thank the Reviewer for this very helpful comment. In this study, we could not systematically control for potential effects of stimulants, such as caffeine or nicotine, on the pupillary reaction. However, while this is a possible limitation of our study, we consider it unlikely that any possible use of stimulants might have significantly affected the results of our study, as the RAPD detects relative differences of the pupillary reaction of both eyes. While stimulants could possibly affect pupillary reactions globally, they appear unlikely to affect relative differences of pupillary reactions of both eyes.

In response to this point, we added the following to the limitation section of the discussion:

„Another possible limitation of our study is that we could not systematically control for potential effects of stimulants, such as caffeine or nicotine, on the pupillary reaction. However, we consider it unlikely that any possible use of stimulants might have significantly affected the results of our study, as the RAPD detects relative differences of the pupillary reaction of both eyes. While stimulants could possibly affect pupillary reactions globally, they appear unlikely to affect relative differences of pupillary reactions of both eyes.” (page 27, line 580-586)

Reviewer’s point:

Line 155: please provide one sample item from the questionnaire and report the Likert scale as well as its psychometric properties based on literature. 

Response: 

According to the Reviewer’s request, we now provide a sample question from the NEI-VFQ and report the Likert scale as well as its psychometric properties based on literature (Mangione et al., 1998) in the revised manuscript:

“The NEI-VFQ is a reliable and valid tool to assess vQoL in a variety of chronic eye conditions. Its strong psychometric properties make it suitable for group-level comparisons in clinical research, offering precise and consistent evaluation of how visual impairment affects the patients’ daily life [22]. To provide an understanding of the items included in the NEI-VFQ, we here present a sample question and the Likert scale of the possible responses. Question: How much difficulty do you have reading street signs or the names of stores? Answers: No difficulty at all, a little difficulty, moderate difficulty, extreme difficulty, stopped doing this because of your eyesight, stopped doing this for other reasons or not interested in doing this." (page 9, lines 179-187)

Furthermore, we specified that the questionnaire utilized in this study was the German adaptation (Franke et al. 1998) of the National Eye Institute-Visual Function Questionnaire:

“We assessed vQoL using the German adaptation [19] of the National Eye Institute-Visual Function Questionnaire (NEI-VFQ) [20].” (page 9, lines 177-178)

Reviewer’s point:

Please also state internal reliability as Cronbach’s alpha.

Response:

We thank the Reviewer for suggesting to use this statistical measure to state internal reliability of the NEI-VFQ. In response, we added the following sentence to the statistics section:

“For internal reliability analysis of the vQoL, we calculated Cronbach’s alpha to assess the internal reliability of the NEI-FVQ.” (page 10, lines 219-221)

Additionally, in the results section, we stated the internal reliability as Cronbach’s alpha by including the following sentence:

“The results of the NEI-VFQ, assessing the vQoL, showed very strong internal reliability (Cronbach’s alpha=0.968).” (page 22, lines 436-437)

Reviewer’s point:

Line 190-191: please discuss the exploratory nature of this study in the introduction.

Response:

We agree with the Reviewer's recommendation to mention the exploratory nature of this cross-sectional observational study. To address this point, we have added the following statement to the introduction:

“In this exploratory study, we systematically evaluated UP for the detection of an RAPD in patients with ON, including a comparison with IVP.” (page 6, lines 103-104)

Reviewer’s point:

Could the authors also provide reference values of the key ON related measures (e.g., PCT) from much larger datasets for readers to compare with the sample in this study?

Response:

To address this point, we performed a literature search, which, however, did not identify large and comprehensive datasets that could provide reference values for the PCT. The largest currently available set of reference values for UP is derived from our own previous study, which included a total study population of n=100. We now refer to these reference values in the following sentence:

“Also, PCT measurements of HC were in line with the reference values (mean, (SD) left eye: 970 (261.6) ms; right eye: 967 (220) ms) for healthy subjects from our previous study [9].” (page 25, lines 520-522)

Reviewer #2 Response to 'correction' document:

Reviewer’s point:

The subheading “Swinging flashlight test” and “Ultrasound pupillometry vs. infrared video pupillometry” in the results section (line 226 and 231 respectively) lack clarity regarding the type of analysis conducted.

Response:

We thank the Reviewer for identifying potential sources of ambiguity. In response to this point, we clarified the respective subheadings:

“Prevalence of an RAPD as assessed by the swinging flashlight test” (page 13, lines 264-265)

and

“Comparative analysis of pupil diameter as assessed by ultrasound pupillometry and infrared video pupillometry” (page 14, lines 270-271)

Reviewer’s point:

Table 2 fails to include the p-value for the correlation coefficient between UP and IVP.

Response: 

We thank the Reviewer for bringing this to our attention. In response, we included the missing p-values in Table 2 (page 15).

Reviewer’s point:

The legend of Figure 4 does not specify the pupillometry method used to measure the pupillary diameter of ON- eyes.

Response:

We thank the Reviewer for raising this point. Of note, Bland-Altman plots are used to graphically illustrate the comparison of two different methods for measuring the same parameter. In this case, the pupil diameter (PD) is being measured by ultrasound pupillometry (UP) and by infrared video pupillometry (IVP). The x-axis reports the difference in PD measurements between the two methods (PD in IVP - PD in UP), while the y-axis shows the mean PD of both methods ((PD in IVP + PD in UP)/2).

To clarify this aspect, we have added both pupillometry assessments in the figure’s caption:

“Fig 4. Bland-Altman plot of pupil diameter as measured by UP and IVP after direct light stimulus in ON- eyes.” (page 17, lines 341-342)

Reviewer’s point:

The legend of Figure 5 does not pr

---

## [Decision Letter · Decision Letter 1]

28 Oct 2024

PONE-D-24-06833R1Ultrasound pupillometry for the detection of a relative afferent pupillary defect (RAPD): systematic evaluation in patients with optic neuritis and comparison with infrared video pupillometryPLOS ONE

Dear Dr. Schmidt,

Thank you for submitting your manuscript to PLOS ONE. After careful consideration, we feel that it has merit but does not fully meet PLOS ONE’s publication criteria as it currently stands. Therefore, we invite you to submit a revised version of the manuscript that addresses the points raised during the review process.

Please address the minor, but significant issues raised by the reviewers regarding publication ethics, in particular dual publication of data, ethics protocols, and proper referencing of literature.

We look forward to receiving your revised manuscript.

Kind regards,

Tudor C. Badea, M.D., M.A., Ph.D.

Academic Editor

PLOS ONE

Journal Requirements:

Reviewers' comments:

Reviewer's Responses to Questions

**Comments to the Author**

1. If the authors have adequately addressed your comments raised in a previous round of review and you feel that this manuscript is now acceptable for publication, you may indicate that here to bypass the “Comments to the Author” section, enter your conflict of interest statement in the “Confidential to Editor” section, and submit your "Accept" recommendation.

Reviewer #1: All comments have been addressed

Reviewer #2: All comments have been addressed

2. Is the manuscript technically sound, and do the data support the conclusions?

Reviewer #1: Yes

Reviewer #2: Yes

3. Has the statistical analysis been performed appropriately and rigorously? 

Reviewer #1: Yes

Reviewer #2: Yes

4. Have the authors made all data underlying the findings in their manuscript fully available?

Reviewer #1: Yes

Reviewer #2: Yes

5. Is the manuscript presented in an intelligible fashion and written in standard English?

Reviewer #1: Yes

Reviewer #2: Yes

6. Review Comments to the Author

Reviewer #1: The authors have thoroughly responded to and tried to address all my comments. Thank you for your effort!

Reviewer #2: Dual Publication: Ensure that the findings and data presented in this manuscript have not been published elsewhere or submitted to another journal simultaneously, as this would constitute dual publication, which violates ethical standards in research publishing.

Research Ethics: The author should confirm that all patient data was handled in accordance with ethical guidelines, including obtaining informed consent and ensuring patient confidentiality.

Publication Ethics: Proper credit should be given through citations to all relevant sources in the literature review. Furthermore, the manuscript should be checked for any potential conflicts of interest or ethical concerns.

7. PLOS authors have the option to publish the peer review history of their article (what does this mean?). If published, this will include your full peer review and any attached files.

Reviewer #1: **Yes: **Xi Yang

Reviewer #2: **Yes: **Priyanka Modi

---

## [Author Response · Author response to Decision Letter 1]

27 Nov 2024

"Ultrasound pupillometry for the detection of a relative afferent pupillary defect (RAPD): systematic evaluation in patients with optic neuritis and comparison with infrared video pupillometry"

Manuscript ID: PONE-D-24-06833R2

Journal Requirement:

Response:

We confirm that all relevant sources are correctly cited in the reference list of the manuscript.

Journal Requirement:

We note that your Data Availability Statement is currently as follows: [Add Data Availability statement here].

Please confirm at this time whether or not your submission contains all raw data required to replicate the results of your study. Authors must share the “minimal data set” for their submission. PLOS defines the minimal data set to consist of the data required to replicate all study findings reported in the article, as well as related metadata and methods.

Response:

We would like to refer to PLOS ONE policy about data availability: ‘All data and related metadata underlying reported findings should be deposited in appropriate public data repositories, unless already provided as part of a submitted article.’

The complete data set was provided as a separate file in the initial submission of the article, see document ‘Data Table’ (Name: Data Table.xlsx) as part of the uploaded Supporting Information in February 2024. This was done in accordance with the journal's data policy. Should this be insufficient, we kindly request that you inform us.

Journal Requirement:

Please include a copy of Table 2 which you refer to in your text on page xx.

Response:

We would like to thank for this remark and provide a revised manuscript with the requested Table 2 entitled ‘Table 2. Infrared video pupillometry vs. ultrasound pupillometry in ON+ and HC eyes.’ (page 14)

Response to Reviewers

Reviewer #1:

Reviewer’s point:

The authors have thoroughly responded to and tried to address all my comments. Thank you for your effort!

Response: 

We appreciate this comment.

Reviewer #2:

Reviewer’s point:

Dual Publication: Ensure that the findings and data presented in this manuscript have not been published elsewhere or submitted to another journal simultaneously, as this would constitute dual publication, which violates ethical standards in research publishing.

Response:

We herewith confirm that the findings and data presented in this manuscript are original and have not been published elsewhere or submitted to another journal simultaneously, as already stated in the original Cover Letter from February 2024. We are fully aware of and adhered to the ethical standards in research publishing, including the guidelines concerning dual publication, and we are committed to maintaining the integrity of the scientific literature.

Reviewer’s point:

Research Ethics: The author should confirm that all patient data was handled in accordance with ethical guidelines, including obtaining informed consent and ensuring patient confidentiality.

Response:

We herewith confirm that all patient data were handled in strict accordance with the ethical guidelines that govern our research. As already stated in the manuscript, written informed consent was obtained from all participants prior to inclusion into the study and patient confidentiality was ensured throughout the study. In response to the Reviewer’s comment, we have amended the respective passage in the Methods Section of the manuscript to mention more explicitly that patient confidentiality was ensured throughout the study: 

“This cross-sectional observational study was approved by the Institutional Review Board of Charité – Universitätsmedizin Berlin (EA1/190/15) and was conducted in accordance with the current applicable ethical guidelines of the Declaration of Helsinki and German law. Written informed consent was obtained from all participants prior to inclusion into the study and patient confidentiality was ensured throughout the study.” (page 6, lines 110-114)

Reviewer’s point:

Publication Ethics: Proper credit should be given through citations to all relevant sources in the literature review. Furthermore, the manuscript should be checked for any potential conflicts of interest or ethical concerns.

Response:

We concur with the Reviewer's comment and thank them for raising this important point. We confirm that all relevant sources have been subjected to a comprehensive review and cited in accordance with the standards set forth in the literature review. In instances where citations were absent, we have duly added them. Furthermore, the manuscript has been meticulously examined for any potential conflicts of interest or ethical concerns, and all necessary disclosures have been included to ensure compliance with the standards set forth in the publication ethics code.

Reviewer’s point:

In the comment regarding page 6, lines 95-98: "The author states that 'UP does not require patient cooperation.' However, there is no discussion about the feasibility of UP in agitated patients. This statement is only partially correct, as UP does not fully eliminate the need for patient cooperation."

Response:

We thank the Reviewer for bringing this to our attention. In response, we included the following additional statement:

“(…) UP does not require active patient cooperation, provided that the patient is able to remain seated or lie quietly during the procedure.” (page 6, lines 95-96)

Reviewer’s point:

In the comment regarding page 25, lines 532-538: "The author has not provided citations to support the literature review in this section."

Response: 

We thank the Reviewer for highlighting this important point. In response, we have incorporated the missing references into the manuscript (Schmidt et al. 2017; Broadway, 2012; Thompson et al. 1981; Beisse et al. 2020).

“UP is an objective and precise measuring tool, providing consistent and reliable measurements [9], whereas the SFLT relies on subjective visual evaluation by the examiner [6, 7]. In the SFLT, results depend on the examiner’s experience and judgment [8].” (page 25, lines 513-516)

---

## [Editor Report · Decision Letter 2]

2 Dec 2024

Ultrasound pupillometry for the detection of a relative afferent pupillary defect (RAPD): systematic evaluation in patients with optic neuritis and comparison with infrared video pupillometry

PONE-D-24-06833R2

Dear Dr. Schmidt,

We’re pleased to inform you that your manuscript has been judged scientifically suitable for publication and will be formally accepted for publication once it meets all outstanding technical requirements.

Kind regards,

Tudor C. Badea, M.D., M.A., Ph.D.

Academic Editor

PLOS ONE
---

## [Editor Report · Acceptance letter]

27 Dec 2024

PONE-D-24-06833R2 

PLOS ONE

Dear Dr. Schmidt, 

I'm pleased to inform you that your manuscript has been deemed suitable for publication in PLOS ONE. Congratulations! Your manuscript is now being handed over to our production team.

Kind regards, 

on behalf of

Dr. Tudor C. Badea 

Academic Editor

PLOS ONE